# Numerical Investigation on Flowability of Pulverized Biomass Using the Swelling Bed Model

Mateusz Przywara [1,*], Regina Przywara [2] and Wojciech Zapała [1,*]

1 Department of Chemical and Process Engineering, Chemical Faculty, Rzeszów University of Technology, Powstańców Warszawy 6, 35-959 Rzeszów, Poland
2 Doctoral School of the Rzeszów, University of Technology, Powstańców Warszawy 12, 35-959 Rzeszów, Poland
* Correspondence: m.przywara@prz.edu.pl (M.P.); ichwz@prz.edu.pl (W.Z.)

**Abstract:** Numerical investigations on the flowability of pulverized biomass are crucial for agriculture, aiding in optimizing biomass use, crop residue management, soil health improvement, and environmental impact mitigation. Rising interest in biomass and conversion processes necessitates deeper property understanding and technological process optimization. Moisture content is a key parameter influencing biomass quality. In this paper, computer simulations of shear tests depending on the moisture content using the discrete element method were carried out and compared with experimental results. An experimental study and modeling for Jenike's direct shearing apparatus was carried out. A swelling bed model was proposed to account for the effect of moisture. The swelling bed model assumed an increase in biomass grain vorticity proportional to the moisture content. The model was solved using the discrete element method (DEM). The model considers the effect of moisture on the values of Young's and Kirchoff's moduli for biomass grains. The model assumed that moisture is not present in surface form, the total amount of moisture is absorbed into the interior of the material grains, and the volume of a single grain increases linearly with an increase in the volume of the absorbed moisture. The tested materials were pulverized sunflower husks, apple pomace, distiller's dried grains with solubles (DDGS), meat and bone meal (MBM), and sawdust. Samples with moisture contents of 0%, 10%, 20%, and 30% were tested. The best agreement of the model with the experimental data was observed for the most absorbent materials in which moisture was not present in surface form, such as apple pomace, DDGS, and sawdust. Research data are important for the proper design of biomass storage, transportation equipment, and utilization as feedstock for bioenergy production or soil enrichment.

**Keywords:** biomass; flowability; discrete element method; DDGS; meat and bone meal; sawdust

## 1. Introduction

Understanding the flow behavior of pulverized biomass is crucial for its efficient utilization in various agricultural applications [1]. Pulverized biomass, such as agricultural residues and energy crops, can be used for energy generation, soil amendment, and animal bedding, among other uses. Efficient handling of pulverized biomass is essential in agricultural operations, particularly in biomass processing plants and bioenergy facilities. The flowability of biomass affects processes such as conveying, storage, and feeding into conversion systems [2] (e.g., gasifiers and fermenters). Biomass flow characteristics influence the performance of conversion processes like pyrolysis, gasification, and combustion [3]. Understanding the behavior of biomass particles within these processes can help optimize the reactor design, enhance the conversion efficiency, and reduce the operating costs. Work is underway to transition to more modern and environmentally friendly technologies that will help change procedures and habits in both production and consumption [4].

Agricultural residues are often left in fields after harvest. Understanding the flowability of pulverized crop residues can inform practices related to residue management,

such as the collection, storage, and utilization as feedstock for bioenergy production or soil enrichment. Incorporating pulverized biomass into soil can improve the soil structure, water retention, and nutrient content. Understanding how biomass particles flow and interact with the soil can aid in developing strategies for soil enrichment and sustainable agriculture practices. Utilizing biomass as a renewable energy source can contribute to reducing greenhouse gas emissions and mitigating climate change. Numerical investigations on biomass flowability can support the development of sustainable agricultural practices that promote biomass utilization and carbon sequestration [5].

Growing interest in renewable energy sources, including biomass, and the growing number of biomass conversion processes make it necessary to design more environmentally friendly processes for its handling, as well as the need to optimize already existing technological processes [6]. The sustainable development of biomass energy within the context of carbon neutrality is a critical aspect of transitioning to a low-carbon and environmentally friendly energy system [5,7].

A number of conversion processes use biomass in powder form as raw material. A homogeneous powder material seen from a distance appears to be a continuum with properties showing similarities to solids, liquids, and gases. When this material is channeled through deflected troughs or walls, it becomes distorted under the influence of a high shear rate, rendering obsolete the continuous mechanical models created on the basis of a solid-like state [8]. Since under such conditions the shear stresses are proportional to the shear rate, it is possible to derive models based on fluid mechanics and kinetic gas theory. The complex behavior exhibited by bulk materials on a macroscopic scale is due to relatively simple but repeatedly applied microscopic interaction laws, which remain constant if atomic conditions do not change quantitatively. This fundamental property can be exploited in direct particle simulation methods such as the discrete element method (DEM) [9].

The discrete element method is one of the most popular numerical methods for determining particle motion trajectories and particle interactions [10]. This method has gained widespread acceptance in many engineering applications where materials are in powdered form. It makes it possible to calculate the physical properties of a large number of objects in free motion. The application of the DEM method is very wide. It makes it possible to simulate the movement of bulk materials by screw and spiral conveyors, both qualitatively and quantitatively [11]. It can be used to design transport equipment and contribute to the clarification and improvement of processes occurring during transport, for example, by simulating transfer stations, which are a sensitive node of belt transport [12]. Simulations make it possible to prevent such transport-related phenomena as jet deflection, splashing, the possibility of blockage formation, and excessive dusting. By associating force magnitudes with grain velocities, it is possible to theoretically predict the locations of excessive loads and take measures to extend the life of the equipment. Computer simulation of grain migration using the DEM method is a helpful tool in the design of vertical bucket elevators and scraper conveyors, as it offers the ability to trace the behavior of almost every particle of material [13]. Sakaguchi and Favier [14], simulating the shearing process of a bulk material in a direct shear apparatus using this computational method, showed that it is a convenient tool for finding connections between micro- and macro-scale interactions. Iwashita and Oda [15] proposed a modified MDEM discrete element method. They assumed that the particle contact region is an area, rather than a contact point, as previously assumed. The replacement of contact points with contact areas introduces the possibility of taking into account rotational motion resistances.

The DEM method is widely used to model the flow ability of powders [16]. The DEM method can be also used when modeling the flowability of moist hygroscopic powders, as in the case of biomass. The presence of moisture can significantly change the properties of powder bed particles [17,18]. Therefore, there is a need for a deeper understanding of the effect of the moisture content on the flow ability of powdered biomass, in which the DEM method can be helpful. The level of the observed difference between experimental

data and modeling can be a reliable measure of the changes in the flowability caused by the moisture content.

In relation to the above considerations, the aim of the study was to develop a model that takes into account the effect of the moisture content on the flow properties of powdered biomass. A swelling bed model was proposed, discussed in detail in the next part of the manuscript. The model allows for performing tests based on changes in the properties of powder materials under the influence of moisture. With a change in moisture, only the input data for the simulation are changed, while the program itself is not changed, resulting in shorter calculation times.

The manuscript is structured as follows: The next section describes the raw materials and analytical methods, as well as the swelling bed model and the DEM calculation method adopted. The following section contains a presentation of the results obtained and their discussion and interpretation. The last section summarizes the results and provides information on future work.

## 2. Materials and Methods

The investigations of the influence of the moisture content on the flowability of pulverized biomass are based on experimental data previously published by the authors of this manuscript [19,20]. The materials selected for the study were as follows: corn dried distiller's grains with solubles (DDGS) (5–600 μm) from the Goświnowice Ethanol Plant, Nysa, Poland; meat and bone meal (MBM) (150–2000 μm) from SARIA Sp. z o.o. Przewrotne, Poland; sunflower husks (180–800 μm) and apple pomace (250–2000 μm) from the Stalowa Wola S.A. Combined Heat and Power Plant, Poland; pine sawdust (10–550 μm), beech sawdust (135–780 μm), oak sawdust (170–720 μm), and a sawdust mixture (60–530 μm) of pine sawdust (60%), beech sawdust (20%), oak sawdust (10%), and sycamore sawdust (10%) from Drewnotech, Rzeszów, Poland. Samples with moisture contents of 0%, 10%, 20%, and 30% were prepared.

An indication of the potential powder flowability in static conditions is the powder yield strength, which was measured with a Jenike shear tester manufactured by CDK Gliwice, Poland. The shear test consisted of placing a sample with a pre-prepared moisture content in a cylindrical chamber consisting of two rings. The chamber was then loaded with a normal force $F_n$ and the movement of one part of the measuring chamber was induced (Figure 1). During the measurement, the tangential force $F_t$ was recorded, the value of which increased. The maximum of the tangential force was recorded. Measurements were repeated for four values of the normal force. The obtained results were plotted on a graph of the dependence of the shear stress $\tau$ on the normal stress $\sigma$ and approximated by a line, the so-called yield locus. The higher the line of the yield locus and the larger its angle of inclination, the worse the flowability of the material.

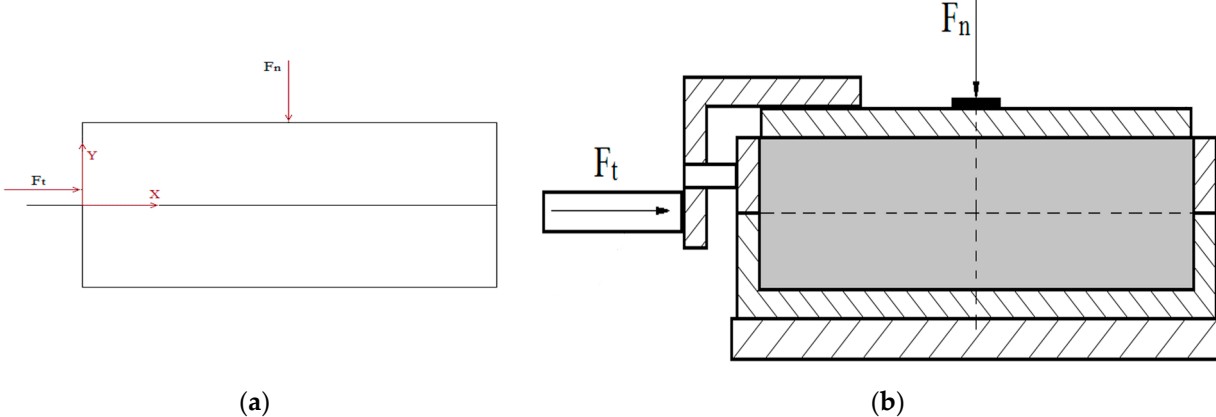

(**a**)  (**b**)

**Figure 1.** (**a**) Jenike's apparatus chamber, created in PFC2D 4.0 software. (**b**) Schematic representation of Jenike's apparatus chamber [19].

Details of how to perform shear strength measurements with a Jenike apparatus and details of the sample preparation and characterization can be found in the authors' previously mentioned published papers [19,20].

The experimental results obtained were compared with modeling results using the DEM method and the swelling bed model described in the following sections of this article.

### 2.1. Calculation Cycle of the DEM Method

The DEM model calculations used Newton's second law of dynamics and a linear particle contact model [9]. In the first step, the boundary conditions were set and the initial positions of the particles were determined. Starting the simulation creates contact points between two particles or between a particle and a wall. For each particle, the contact force resulting from the specified contact model is calculated. Then, using Newton's second law of dynamics, the displacements of the particles are determined based on the resultant force acting on them. It is assumed that, in the direction normal to the contact surface, there is a visco-elastic interaction between the grains of the medium, and, in the tangential direction, there is a visco-elastic-plastic interaction.

The time step is assumed to be small enough so that the interactions between the particles do not transfer further than to a neighboring particle. The time step is one of the most important parameters ensuring the stability of the simulation while limiting its execution time.

The calculation cycle was repeated until the assumed number of time steps was reached or the equilibrium state of the system was reached.

The first step was to calculate the force $F_i$ at contact point $i$, consisting of the normal factor $F_i^n$, working along the line connecting the centers of the contacting particles, and the tangential factor $F_i^s$, working in the plane of contact, as follows:

$$F_i = F_i^n + F_i^s \tag{1}$$

The next step was to calculate the normal factor component of $F_i^n$, as follows:

$$F_i^n = K^n + U^n \tag{2}$$

where:

$K^n$—the coefficient of the linear contact stiffness in the normal direction, determined based on the current contact model;

$U^n$—the relative displacement of the point of contact of the particles, along the line connecting their centers.

The tangential factor component $F_i^s$ is then calculated as follows:

$$F_i^s = F_i^{s-1} + \Delta F_i^s \leq \mu F_i^n \tag{3}$$

where:

$F_i^{s-1}$—the tangential factor of the force from the previous time step.

The condition for calculating the tangential component $F_i^s$ is not to exceed the product of the normal component $F_i^n$ and the friction coefficient $\mu$. Exceeding the mentioned value causes the mutual slippage of the particles. The increment of force in the tangential direction $\Delta F_i^s$ is calculated as follows:

$$\Delta F_i^s = -K^s + \Delta V^s \tag{4}$$

where:

$K^s$—the coefficient of the contact stiffness in the tangential direction;

$V^s$—the relative shear velocity at the point of contact.

Adding up the forces acting on a single particle at all points of contact for a given calculation cycle yields the resultant force $F$, by which the acceleration acting on a single particle is calculated according to Newton's second law of dynamics. Assuming that,

within the time step $\Delta t$, the acceleration is constant, the new particle position is calculated as follows:

$$x^{(t+\Delta t)} = x^{(t)} + \dot{x}^{(t-\frac{\Delta t}{2})} \Delta t \tag{5}$$

where:

$x^{(t)}$—the previous position of the particle;

$\dot{x}^{(t-\frac{\Delta t}{2})}$—the velocity of the particle at the previous time step;

$\Delta t$—the time step.

The velocity of a particle $\dot{x}^{(t+\frac{\Delta t}{2})}$ with mass $m$ is given by the following equation:

$$\dot{x}^{(t+\frac{\Delta t}{2})} = \dot{x}^{(t-\frac{\Delta t}{2})} + \left(\frac{F}{m} + g\right)\Delta t \tag{6}$$

where:

$\dot{x}^{(t-\frac{\Delta t}{2})}$—the velocity of the particle in the previous time step;

$F$—the resultant force acting on the particle;

$m$—the mass of the particle;

$g$—the ground acceleration.

Particle Contact Model

There are two types of contact models, taking into account the character of the dependence of contact forces $F_i$ on the deformation $\alpha$: linear and nonlinear models [21]. The general form of the force–strain dependence in the normal direction is as follows:

$$F_i = K\alpha^\eta \tag{7}$$

For linear models, $\eta = 1$, and for nonlinear models, $\eta \neq 1$, $K$ is the stiffness coefficient taking different values depending on the type of material.

For the linear model, the contact stiffness coefficient in the normal direction $K^n$ is calculated based on the stiffness moduli of the contacting particles as follows:

$$K^n = \frac{k_n^{[A]} \cdot k_n^{[B]}}{k_n^{[A]} + k_n^{[B]}} \tag{8}$$

The particle stiffness coefficient in the tangential direction $k^S$ is calculated as follows:

$$k^s = \frac{k_s^{[A]} \cdot k_s^{[B]}}{k_s^{[A]} + k_s^{[B]}} \tag{9}$$

The superscripts A and B represent two particles in contact. For the linear model, the modulus of stiffness of the particles in the normal direction $k^n$ is equal to the coefficient of stiffness of contact in the normal direction $K^n$, as follows:

$$k^n \equiv \frac{dF^n}{dU^n} = \frac{d(K^n \cdot U^n)}{dU^n} = K^n \tag{10}$$

where:

$F^n$—the normal component of the contact force;

$U^n$—the relative displacement of the point of contact of the particles along the line connecting their centers.

### 2.2. Numerical Investigation Metodology

Simulations were performed using PFC2D software from Itasca Consulting Group, Barrel House, MN, USA. In the simulations conducted, a linear contact model was used. At the first stage of the simulation, the main part of the Jenike apparatus, i.e., the measuring

chamber (Figure 1), was recreated. The height of the chamber was H = 30 mm and its width was D = 60 mm, and the chamber was divided into two elements with a height of 15 mm each; the movement speed of the lower moveable part of the chamber was established equal to $v = 4.65 \times 10^{-5}$ m/s. The loading of the bed with normal stress σn was realized with the upper wall of the chamber moving in the negative direction to the Y axis. The movement of the lower part of the chamber was carried out in the positive direction of the X axis.

Simulation of the direct shear test included the following steps:

1.  Filling the test chamber with material of the specified parameters;
2.  Consolidation of a bed of powder material, consisting of loading the bed through the top wall of the measurement chamber, carried out until a preset value of the consolidation stress $\sigma_n$ was obtained. In the course of conducting the measurement, this stress was kept constant through the use of the so-called servomechanism;
3.  The preliminary shearing of the bed of the test material forced by the movement of the lower part of the measuring chamber, at the preset consolidating stress $\sigma_n$, carried out until the measured normal stress reaches the maximum value;
4.  The shear test, conducted for four progressively decreasing values of normal stress and recording maximum values of tangential stress.

Swelling Bed Model

In order to take into account the effect of the moisture content on the properties of the tested materials, a swelling bed model was proposed. Due to the fact that the tested materials were of plant origin, it was assumed that the moisture content in the tested range (from 0% to 30%) was entirely absorbed into the grain of the material and did not occur on the surface. The validity of the above assumption was confirmed by the aforementioned typical saturation point values for wood.

The swelling bed model was based on the following assumptions:

1.  Moisture is not present in surface form; the total amount of moisture is absorbed into the interior of the material grains;
2.  The volume of a single grain of material increases linearly with an increase in the volume of the absorbed moisture;
3.  The density of the material changes proportionally to the changes in moisture content;
4.  The values of the modulus of elasticity in the tangential and normal directions decrease linearly with the increasing moisture content.

Measurements were conducted at normal stress values that were the same as in laboratory tests; for most materials, the consolidating stress was $\sigma_n$ = 22.2 kPa; the four normal stress values were $\sigma_1$ = 20.8 kPa, $\sigma_2$ = 18.4 kPa, $\sigma_3$ = 13.0 kPa, and $\sigma_4$ = 5.4 kPa. The exceptions were sunflower hulls and apple pomace, for which the consolidation stress was assumed equal to $\sigma_n$ = 80.3 kPa, and the normal stresses were $\sigma_{1n}$ = 72.5 kPa, $\sigma_2$ = 54.3 kPa, $\sigma_3$ = 33.2 kPa, and $\sigma_4$ = 13.0 kPa. The detail of the selection of the mentioned parameters were described in a previous paper [19].

The input data for the simulation can be divided into two groups: data for the Jenike apparatus model constructed with WALLS (walls) objects, and data for the tested material with BALL (particle) objects. The assumed input data for the Jenike apparatus model are summarized in Table 1.

**Table 1.** Input data for modeling of the Jenike apparatus.

| Input Data | Symbol | Value |
| --- | --- | --- |
| Modulus of elasticity of wall in normal direction | wall_kn | $205 \times 10^9$ [Pa] |
| Modulus of elasticity of wall in tangential direction | wall_ks | $80 \times 10^9$ [Pa] |
| Consolidating stress | swall_target_str | $2.22 \times 10^4$ [Pa] |
| Speed of the lower part of the measuring chamber | wall_v | $4.65 \times 10^{-5}$ [m/s] |

The discrete element method used by the PFC2D program requires knowledge of the moduli of elasticity of the walls and particles, as well as the friction coefficients and other material data, as in Table 2. The values of the moduli of elasticity of the particles in the normal and tangential directions were assumed to be equal to Young's and Kirchoff's moduli, respectively, similarly for the walls, using data as for steel. As noted by Simons et al. [22], Young's modulus and the static and rolling friction coefficients are the most influential particle parameters in the observed tangential shear stress analysis conducted using Schulze ring shear tester. The values of the moduli of elasticity for the biomass were taken from the literature [23–25]. As reported in the aforementioned literature, the values of the moduli of elasticity decrease with increasing moisture content. Based on the work of [24,25], it was assumed that a 10% increase in the moisture content causes a 50% decrease in the value of the elastic moduli. Table 2 presents an example of input data set for oak sawdust with a 0% moisture content.

**Table 2.** Input data for oak sawdust with a 0% moisture content.

| Input Data | Symbol | Value |
|---|---|---|
| Modulus of elasticity of wall in normal direction | ball_kn | $10 \times 10^9$ [Pa] |
| Modulus of elasticity of wall in tangential direction | ball_ks | $4 \times 10^9$ [Pa] |
| Density | ball_dens | 800 [kg/m$^3$] |
| Minimum particle radius | r_lo | $8.55 \times 10^{-5}$ [m] |
| Ratio of maximum particle radius to minimum particle radius | r_rat | 4.22 [-] |
| Porosity of the bed | poros | 0.35 [-] |
| Friction coefficient between particles | ball_fric | 0.747 [-] |
| Friction coefficient between the particle and the wall | wall_fric | 0.2 [-] |

Particle parameters, such as the minimum particle radius, the ratio of the maximum particle radius to the minimum radius, and the friction coefficients between particles, were calculated based on the data obtained experimentally. The minimum particle radius $r\_lo$ was calculated as follows:

$$r\_lo = \frac{d(0,1)}{2},\tag{11}$$

The ratio of the maximum particle radius to the minimum particle radius $r\_rat$ was calculated as follows:

$$r\_rat = \frac{\frac{d(0,9)}{2}}{\frac{d(0,1)}{2}},\tag{12}$$

where $d(0.X)$ is the volume diameter where X% of the particles is below X%. The diameters were evaluated using a Mastersizer 2000 (Malvern, UK). The friction coefficient between particles was calculated as follows:

$$f = tg\varphi,\tag{13}$$

where $\varphi$ is the angle of internal friction, measured experimentally with a Jenike apparatus for materials with different moisture contents.

The PFC2D program calculates the number of simulated particles based on the grain size and assumed porosity. In order to eliminate the effect of porosity, a constant value was assumed without changing it with a change in the moisture content. The values of the particle-wall friction coefficient and density were assumed based on the works of [24,25].

## 3. Results and Discussion

The following section shows the effect of the moisture content on the course of the yield loci of the modeled materials and a comparison with the experimental results. Figure 2 shows the chamber of Jenike's apparatus filled with DDGS; this is the first stage of the simulation. In the next stage, the bed is loaded with normal stress (Figure 3); after a

constant value of normal stress is obtained, the movement of the lower part of the test chamber begins.

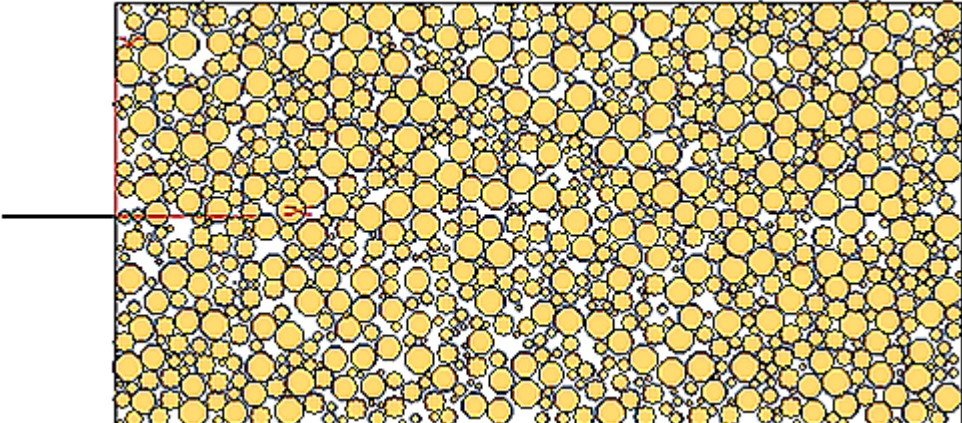

**Figure 2.** The measuring chamber of the Jenike apparatus with the generated DDGS bed with a 0% moisture content.

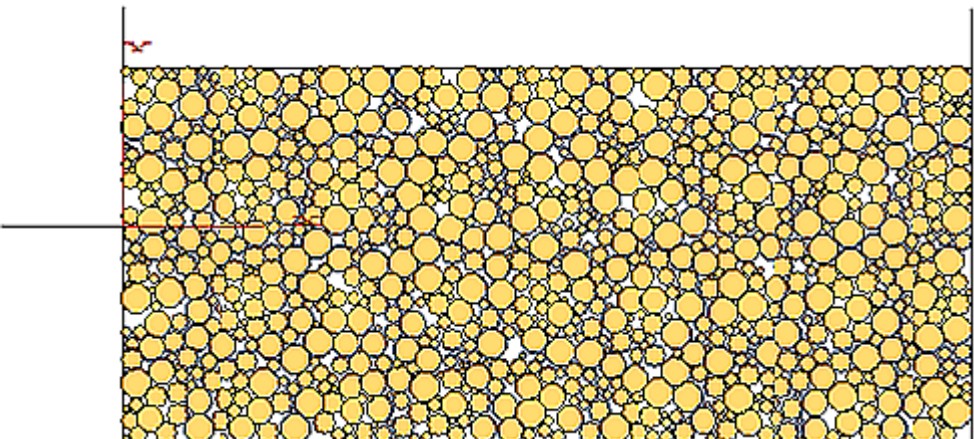

**Figure 3.** The measuring chamber of the Jenike apparatus with the generated DDGS bed with a 0% moisture content, loaded with a normal stress $\sigma_n$ = 22.2 kPa.

The displacement vectors in Figures 4 and 5 demonstrate the greater packing of the particles caused by their greater deformation in the case of the material with a 30% moisture content. An increase in the moisture content of the material causes an increase in the stress. The particles of the powder material move larger distances in this case during consolidation. For the moist material, a reduction in the bed volume was observed. Based on the visualization, it is only possible to make a rough assessment of the change in the stress with a change in the moisture content; the basis for a detailed analysis is the calculated values of the maximum shear stresses.

The first stage of the simulation is to load the bed with a normal stress $\sigma_n$. The software adjusts the position of the top wall of the chamber so as to obtain the set value of the vertical stress, and then keeps it constant. Figure 6 shows the dependence of normal stresses $\sigma_n$ from time $t$ for dry material and material with a 30% moisture content. As can be seen, the set stress value is obtained at a similar time, and the increase in moisture content does not have a major impact on the process of loading the bed.

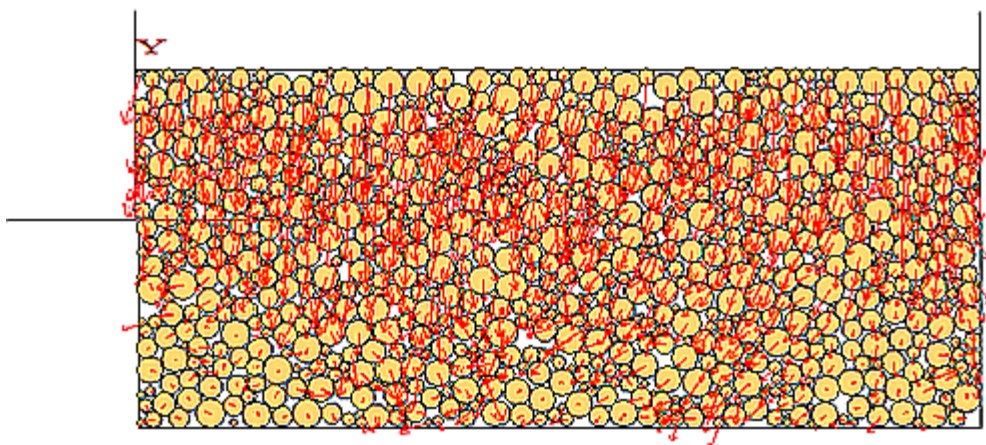

**Figure 4.** Displacement vector distributions for the shear test of a DDGS bed with a 0% moisture content.

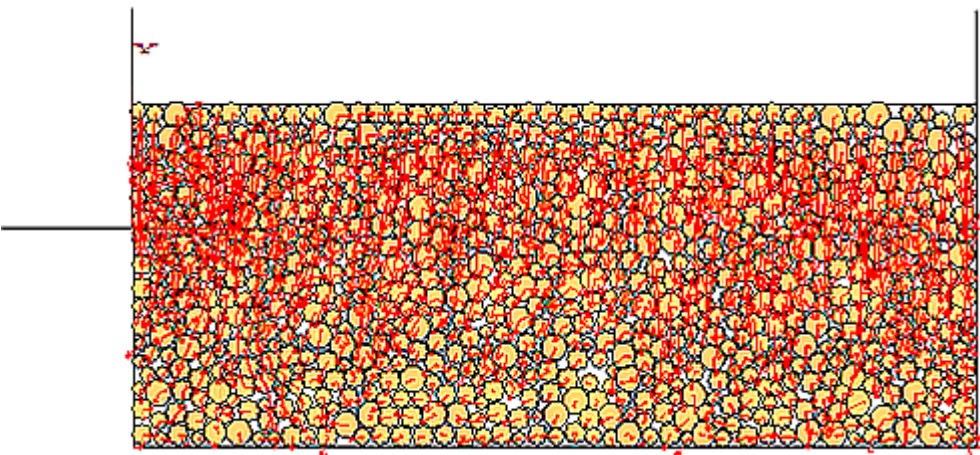

**Figure 5.** Displacement vector distributions for the shear test of a DDGS bed with a 30% moisture content.

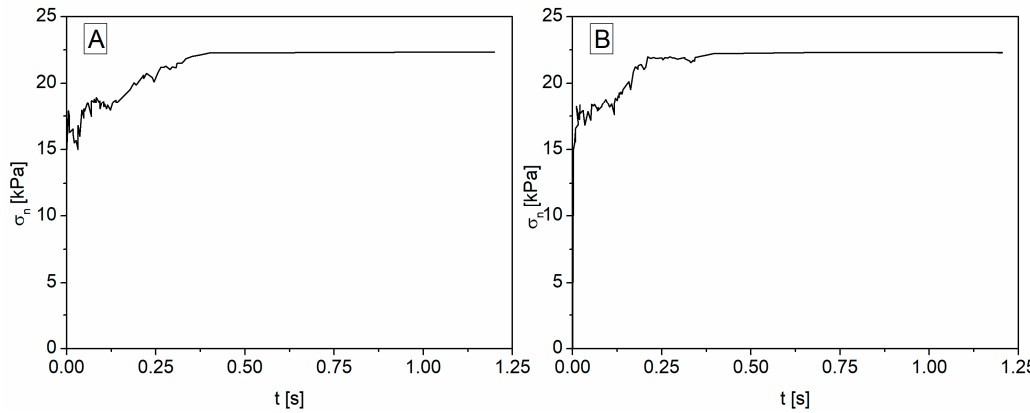

**Figure 6.** Dependence of normal stress $\sigma_n$ as a function of time for the DDGS with a 0% (**A**) and 30% moisture content (**B**).

Figure 7A–D show the changes in the tangential stress $\tau$ as a function of time for dry material and material with a 30% moisture content. Tangential stress increases as the shearing process progresses; initially, it is a sharp increase, then the stress value stabilizes, and when the maximum value is reached, the program stops the process. Such a course of changes in the tangential stress was also observed in the case of the experimental studies. Therefore, it can be concluded that the assumptions made are correct and that the DEM method and the linear particle collision model are useful in the analysis of fragmented

systems. No significant differences were observed in the course of tangential stress changes for individual materials. Changes in the moisture content also do not significantly affect the course of tangential stress changes; the only noticeable difference is the maximum value of the tangential stress, which, as in most cases of laboratory tests, increases with increasing moisture content.

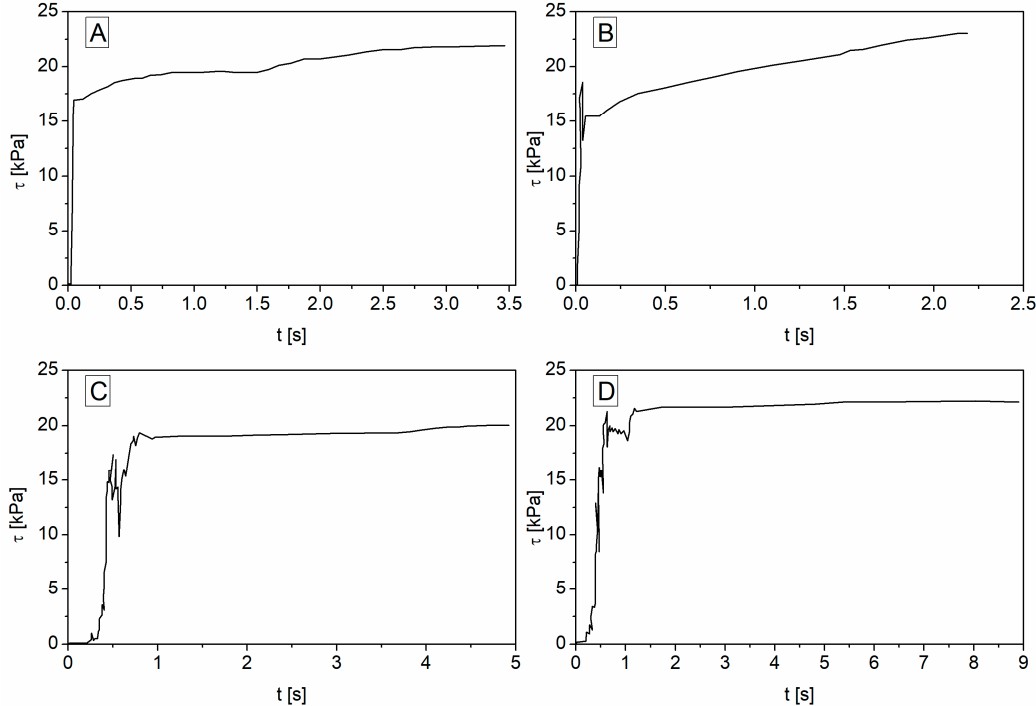

**Figure 7.** Dependence of tangential stress $\tau$ as a function of time for the DDGS with a 0% (**A**) and 30% moisture content (**B**), and for beech sawdust with a 0% (**C**) and 30% moisture content (**D**) for a normal stress $\sigma_n$ of 22.2 kPa.

The results of the study of the effect of the moisture content on the mechanical characteristics were presented similarly to the experimental studies [19] in the form of a flow condition, i.e., the dependence of the tangential stress $\tau$, that is, the strength of the deposit, on the normal load $\sigma_n$, i.e., the degree of consolidation of the powder material deposit.

The material data used for the simulations and the swelling bed model adopted provided satisfactory results.

As a result of the numerical tests, the values of shear stresses comparable to those obtained for laboratory tests were obtained (Figures 8–15). An analogous effect of the moisture content was observed as in the case of laboratory work. An increase in the moisture content increases the strength of the material and increased the value of the force required to achieve the same deformation. The regular and systematic increase in the tangential stresses is particularly evident in the range of higher normal stresses. The lower the normal stresses, the less the moisture has an effect on the values of the tangential stresses. Differences were observed depending on the origin of the sawdust. The best compliance was obtained with pine sawdust (Figure 8). These are sawdusts with the smallest values of elastic moduli, so they are the most susceptible to deformation. In addition, they absorb moisture very well. In the case of beech (Figure 9) and oak sawdust (Figure 10), greater discrepancies were observed between the model and the experiment. This may be due to the higher values of the modulus of elasticity and the lower absorptivity of these materials. In this case, the swelling bed model may not be sufficient. An analogous increase in shear stresses caused by an increase in moisture content was observed for all types of sawdust tested separately. The sawdust mixture shows less susceptibility to the influence of the moisture content (Figure 11). A trend was also observed for the deposit's liquidity to increase after exceeding a certain moisture content, manifested by the location

of the line for the moist material below the line for the dry material, especially in the range of high values of normal stress. This may be due to the greater fineness of the bed of the sawdust powder mixture. The model assumed a linear increase in the grain size with the increasing moisture content, but, in this case, it was insignificant, which may have inadequately accounted for the effect of moisture.

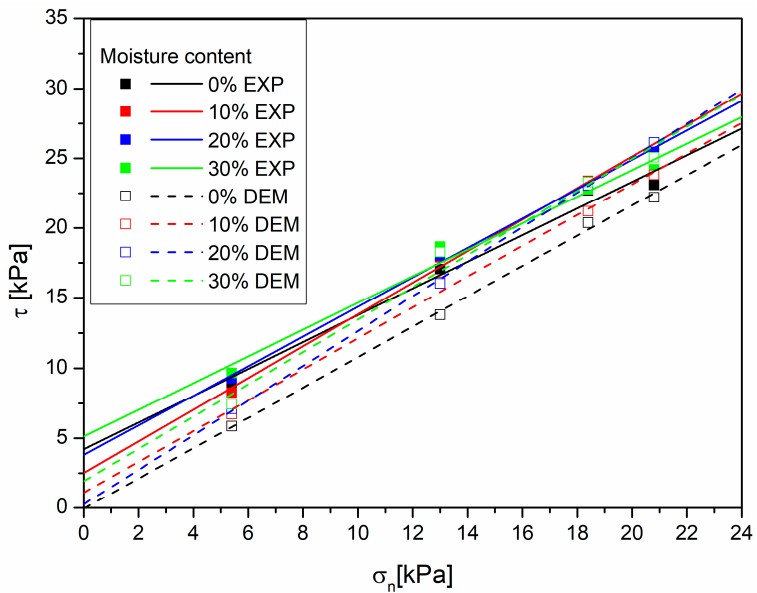

**Figure 8.** Effect of the moisture content on the yield loci for pine sawdust; experimental results—continuous lines and modeling results—dashed lines.

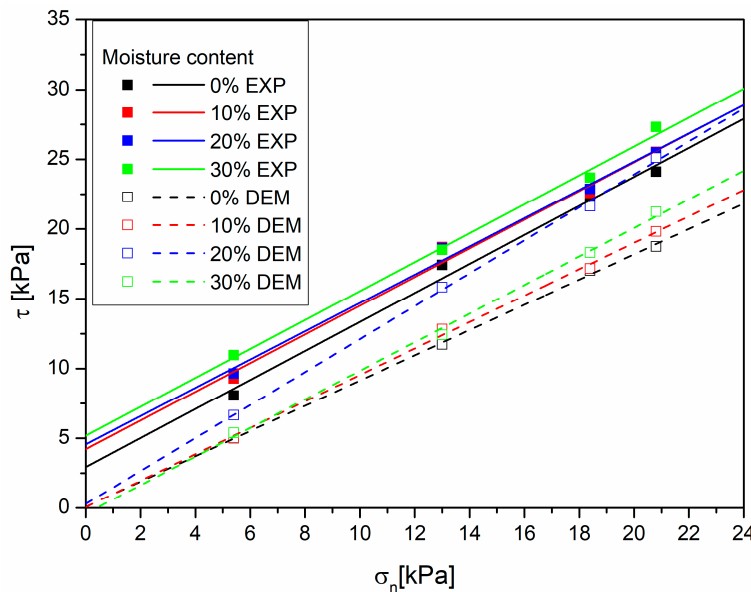

**Figure 9.** Effect of moisture content on yield loci for beech sawdust; experimental results—continuous lines and modeling results—dashed lines.

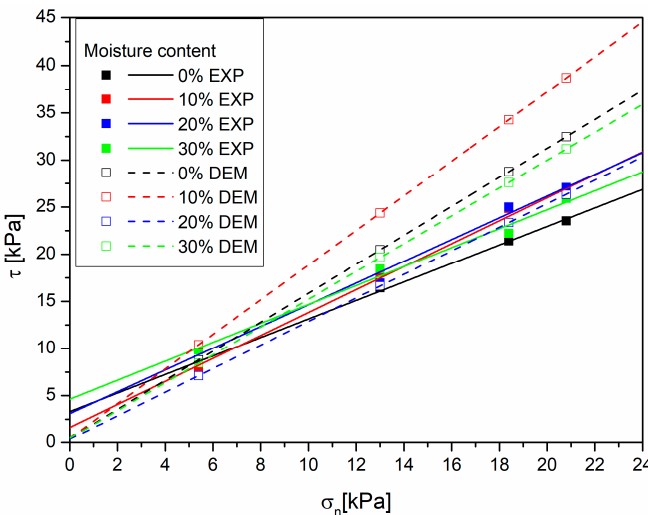

**Figure 10.** Effect of moisture content on yield loci for oak sawdust; experimental results—continuous lines and modeling results—dashed lines.

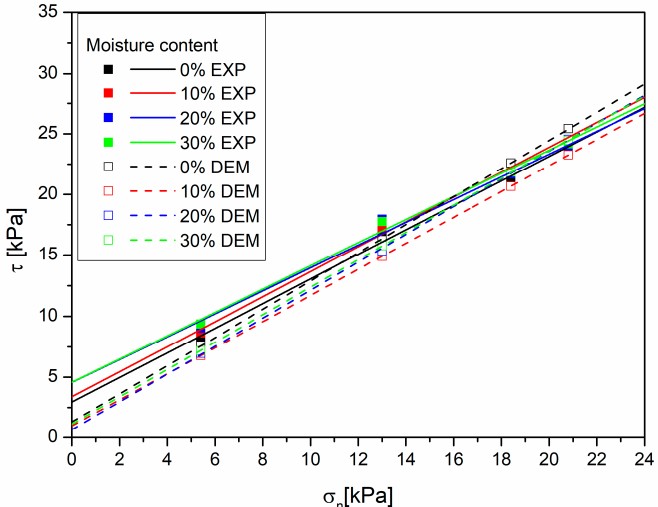

**Figure 11.** Effect of moisture content on yield loci for sawdust mixture; experimental results—continuous lines and modeling results—dashed lines.

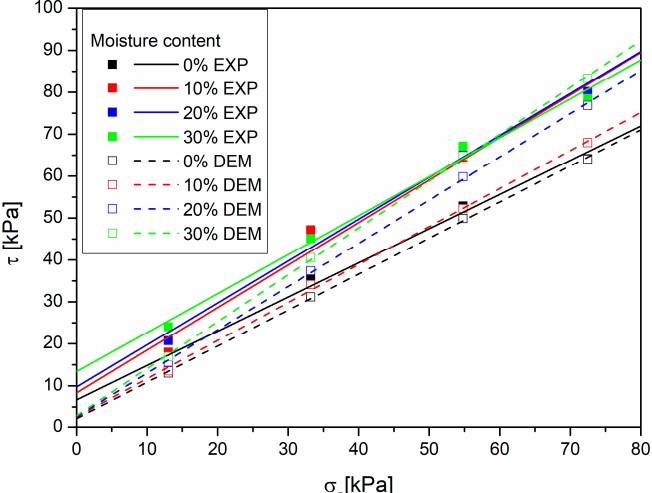

**Figure 12.** Effect of moisture content on yield loci for apple pomace; experimental results—continuous lines and modeling results—dashed lines.

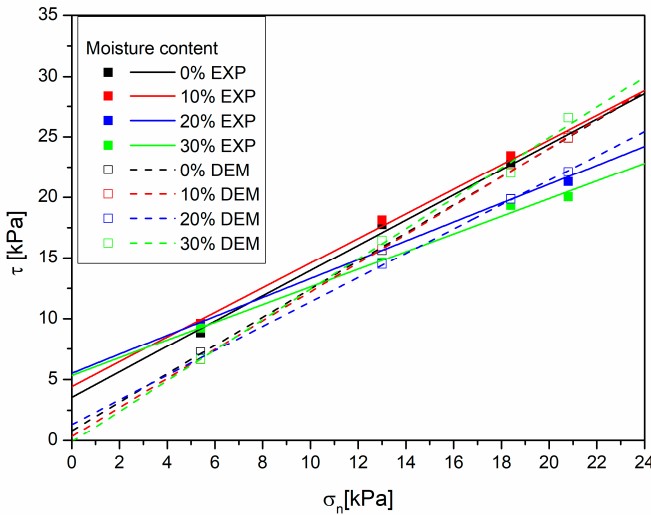

**Figure 13.** Effect of moisture content on yield loci for meat and bone meal; experimental results—continuous lines and modeling results—dashed lines.

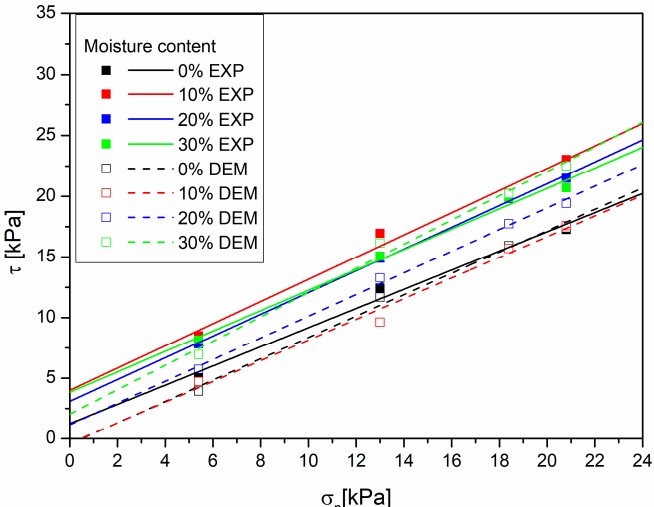

**Figure 14.** Effect of moisture content on yield loci for DDGS; experimental results—continuous lines and modeling results—dashed lines.

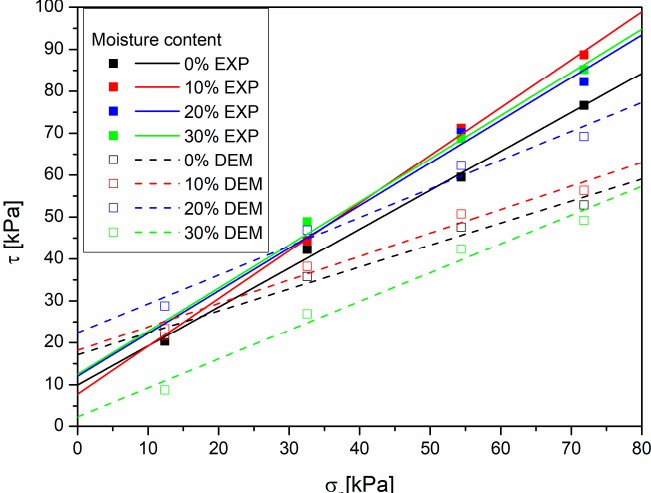

**Figure 15.** Effect of moisture content on yield loci for sunflower husks; experimental results—continuous lines and modeling results—dashed lines.

The effect of the moisture content on the rheology of the bulk powder through capillary cohesion forces is well known and has been described and introduced into classical rheology [26], but only for materials resistant to water sorption and internal diffusion. Hygroscopic materials of biological origin, such as biomass, are problematic in modeling the effect of moisture on flowability, and changes in their properties under the influence of moisture are also difficult to describe. Hygroscopic materials of biological origin at a low water content absorb water and swell up to the saturation point. Crossing the saturation point means starting the process of water adsorption and the formation of liquid bridges. The saturation point is dependent on the properties of the materials. In order to explain such phenomena in the rheology of a powder bed below the saturation point, a swelling bed model has been proposed. Constant values of the friction and damping coefficients were assumed in the calculations. Literature results [27] for semolina indicate that the friction coefficients and damping coefficients of the biological powder particles are not constant, and that they are significantly reduced at a higher moisture content in the powder bed over the entire shear rate range due to the modification of the external surface properties of the particles during bed loading. Under the influence of moisture, the particles become more flexible and deform more easily, changing the surface area. The contact surface grows, causing an increase in friction between particles and an increase in shear stress. The values of the friction coefficients are often adjusted during DEM simulations to match the experimental data [27]. They can be considered quantitative measures of a material's rheological properties.

Similar results were obtained for most of the other biomass samples. Most of the materials studied showed an increase in the shear stress with an increasing moisture content. The best agreement of the model with the experiment was obtained for apple pomace (Figure 12). This may be due to the hygroscopic properties of this material and its good water absorption; in such a case, the swelling bed model gives good results.

In the case of meat and bone meal, satisfactory agreement of the model with experimental data was obtained for higher values of normal stresses (Figure 13). Laboratory tests indicated the appearance of the so-called lubrication effect, i.e., the occurrence of an increase in the fluidity of the bed caused by an increase in the moisture content. Computer simulations showed a regular increase in the shear stress, and thus a decrease in the fluidity caused by an increase in the proportion of moisture in the material. In the case of this material, the presence of moisture was observed on the surface of the crushed bones, and the swelling bed model assumed that the moisture in surface form was not present, and was absorbed entirely into the grains. In this case, the swelling bed method is not sufficient to correctly model this type of material, in which the moisture is also present on the surface of the grains. For the DDGS (Figure 14), a good model agreement was observed, and the bed of this material shows good hygroscopic properties. Some limitations of the DEM method and the swelling bed model can be evidenced by obtaining the negative cohesion values for the DDGS and beech sawdust.

The results obtained for sunflower husks (Figure 15) show the lowest agreement with the experimental data over the entire range of normal stresses. For a more accurate description of the occurring dependencies, additional parameters such as the reduction in Young's modulus with the change in stresses would have to be taken into account [28].

## 4. Conclusions

The results obtained show the accuracy of the proposed swelling bed model and the correctness of the assumptions made.

Based on the results of the experimental investigations, it can be concluded that the research hypotheses tend to be confirmed and the concept seems to be industrially promising. The swelling bed model shows good results for phytomass, for which moisture is absorbed completely into the grain interior of the pulverized bed. The best results were obtained for hygroscopic materials in which surface moisture does not occur, such as apple pomace, DDGS, and pine sawdust.

A limitation of the method is that it does not account for surface moisture, making it less suitable for modeling non-porous materials. One such material is meat and bone meal. The surface moisture provided lubrication of the material layers and reduced stresses. The proposed swelling bed model assumed the absence of surface moisture, all of which was absorbed into the grains, hence the discrepancies between the numerical and experimental methods. Based on the resulting differences, it can be concluded that the lubrication effect was rightly linked to the appearance of moisture in the form of a thin surface layer, the absence of which in the numerical calculations did not cause an increase in the flowability of the powder, as was the case in the laboratory studies.

Research data are important for the proper design of equipment for storing and transporting biomass and using it as a feedstock for bioenergy production or soil enrichment.

Future work will focus on taking into account surface moisture and expanding the swelling bed model to include a model of the deformation of hydrated powders under stress.

**Author Contributions:** Conceptualization, M.P.; methodology, M.P.; software, R.P.; validation, M.P. and W.Z.; formal analysis, M.P.; investigation, M.P.; resources, R.P. and W.Z.; data curation, W.Z.; writing—original draft preparation, M.P.; writing—review and editing, M.P.; visualization, M.P. and R.P.; supervision, W.Z.; project administration, M.P. All authors have read and agreed to the published version of the manuscript.

**Funding:** This research received no external funding.

**Institutional Review Board Statement:** Not applicable.

**Informed Consent Statement:** Not applicable.

**Data Availability Statement:** The data that support the findings of this study are available from the corresponding author upon reasonable request.

**Conflicts of Interest:** The authors declare no conflicts of interest.

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
