# Peer review of "Numerical Investigation on Flowability of Pulverized Biomass Using the Swelling Bed Model"

_agriengineering, doi:10.3390/agriengineering6020078_

Round 1
Reviewer 1 Report
Comments and Suggestions for Authors
Comments can be found in the attached file.

English of the manuscript is fine to me.
Reviewer 2 Report
Comments and Suggestions for Authors
The authors present an interesting paper on pulverized biomass using the swelling bed model. The work brings new aspects to the mechanics of bulk materials and biomass. The only downside , in my opinion, is the use of 2D modeling instead of 3D.
Please comment on this. Why they used a simpler one.
Reviewer 3 Report
Comments and Suggestions for Authors
Dear author(s),
there are some inspiring insights thorough the manuscript and I tend to agree on its publication. However, there are few points that can be quickly addressed to improve its overall communication:
Title:
1/ clearly indicate what was firstly revealed by this "Numerical investigation..."
Abstract:
2/ strictly follow the established schema of writing academic Abstract: A/ introduction (urgency and significance of the research hypothesis); B/ principles of the methods used + key results; C/ conclusions (commercial and environmental impacts)
3/ "DEM" = (DEM)
4/ where is the breakthrough? clearly indicate who (and how) will benefit from these revelations, quantify the industrial importance of your work (preferably in financial terms)
Introduction:
5/ please understand that "biomass" is not just phytomass
6/ remind our readers with the latest trend in the fields you are mentioning, refer to papers "Use of biogas plants on a national and international scale" and "Analysis of Czech Subsidies for Solid Biofuels"
7/ remove all clusters of references to avoid reference overkill (prefer only 1 reference per sentence to support 1 claim)
8/ as far as "renewables" are concerned it needs to be reminded that the cost competitiveness is the main bottleneck, refer to paper "Techno-economic identification of production factors threatening the competitiveness of algae biodiesel"
9/ make sure that this chapter fully introduces any reader into to the topic, explain all the terms, units, abbreviations, Latin and Greek letters, and the whole context that is necessary for anyone (including experts from other disciplines) to understand the following chapters
10/ go straight to the point and more in depth, write more technically (always provide corresponding numbers), significantly condensate all the text by reducing ballast phrases and cliché
11/ remind our readers that the level of disintegration plays a critical role, make sure the urgency and significance of the research hypothesis was justified
Materials and Methods:
12/ increase the professionalism of the manuscript, remove all the "we" and "our"
13/ the method must be presented in such a way that it can be reproduced anytime, by anyone and anywhere (do not create obstacles like referring to specific location, do not indicate dependence on the time of the experiment etc.), please understand that the methodology must be described in a completely unambiguous way that does not allow for multiple interpretations (everyone who reads this chapter should get very precise instructions on how to repeat your procedure to achieve exactly the same results)
14/ each material/reactant and apparatus used needs to be presented in detail (serial number, setup, process parameters, manufacturer, country of origin, purity etc.)
15/ do not ignore (economic) reality, consider providing analysis on material and energy flows that would allow you to consider financial sustainability
16/ check Figure 1: there seems to be some technical issue in the PDF version of the manuscript
Results and Discussion:
17/ each Tab. and Fig. should be provided with caption that describes A/ what can be seen and B/ how is this relevant to the research hypothesis
18/ show more self-criticism to your work (what are the weaknesses of the methods used? are the lessons learned transferable to other fields?)
19/ remind our readers that AI can significantly help solve multifactorial techno-economic problems and refer to papers "Artificial intelligencebased predictive maintenance, time-sensitive networking, and big data-driven algorithmic decision-making in the economics of Industrial Internet of Things" and "The impact of corporate reputation and social media engagement on the sustainability of SMEs: Perceptions of top managers and the owners"
20/ compare your results in more depth with the existing literature, identify the main deviations and try to explain the mechanisms by which they may have been caused
21/ propose some improvements and direction for future research
22/ reveal the main driving mechanisms of your results, provide deeper synthesis and reveal some more original/significant findings
23/ reduce the number of Figures (5 is usually optimum)
Conclusions:
24/ do not repeat what was investigated, do not repeat your methods and results again and again, please understand that the Conclusion chapter is not a summary of your work (such as Abstract), present only original, generalized and industrially significant revelations that have the potential to expand the horizon of human knowledge (higher level of generalization is mandatory)
25/ clearly indicate whether the research hypotheses tends to be confirmed or not and whether the concept seems to be industrially promising (economically sustainable)
Round 2
Reviewer 1 Report
Comments and Suggestions for Authors
I do not have further comments.